# Tumor-Associated Macrophages in Glioblastoma: Mechanisms of Tumor Progression and Therapeutic Strategies

**DOI:** 10.3390/cells14181458

**Published:** 2025-09-18

**Authors:** Jianan Chen, Qiong Wu, Anders E. Berglund, Robert J. Macaulay, James J. Mulé, Arnold B. Etame

**Affiliations:** 1Department of Neuro-Oncology, H. Lee Moffitt Cancer Center and Research Institute, Tampa, FL 33612, USA; jianan.chen@moffitt.org (J.C.); qiong.wu@moffitt.org (Q.W.); 2Department of Quantitative Health Sciences, Division of Computational Biology, Mayo Clinic, 4500 San Pablo Road South, Jacksonville, FL 32224, USA; berglund.anders@mayo.edu; 3Departments of Anatomic Pathology, H. Lee Moffitt Cancer Center and Research Institute, 12902 Magnolia Drive, Tampa, FL 33612, USA; robert.macaulay@moffitt.org; 4Department of Immunology, H. Lee Moffitt Cancer Center and Research Institute, 12902 Magnolia Drive, Tampa, FL 33612, USA; james.mule@moffitt.org

**Keywords:** glioblastoma, tumor-associated macrophages, immunosuppressive environment, immunotherapy, immune evasion

## Abstract

Glioblastoma (GBM) is an aggressive brain tumor with a highly immunosuppressive microenvironment that promotes tumor progression and therapy resistance. Tumor-associated macrophages (TAMs), comprising up to 50% of the tumor mass, are recruited via chemokine axes such as CCL2/CCR2, CX3CL1/CX3CR1, and CXCL12/CXCR4 and adopt an M2-like immunosuppressive phenotype, facilitating immune escape and angiogenesis. Key signaling pathways, including CSF1R, STAT3, NF-κB, PI3K/Akt, and HIF-1α, regulate TAM function, making them promising therapeutic targets. Strategies such as TAM depletion, reprogramming, and immune checkpoint blockade (PD-1/PD-L1, and CD47-SIRPα) have shown potential in preclinical models. Emerging approaches, including CAR-macrophage (CAR-M) therapy, nanotechnology-based drug delivery, and exosome-mediated modulation, offer new avenues for intervention. However, clinical translation remains challenging due to GBM’s heterogeneity and adaptive resistance mechanisms. Future research should integrate multi-omics profiling and AI-driven drug discovery to refine TAM-targeted therapies and improve patient outcomes. This review provides a comprehensive analysis of TAM-mediated immune regulation in GBM and explores evolving therapeutic strategies aimed at overcoming its treatment barriers.

## 1. Introduction

Glioblastoma (GBM) is the most aggressive primary malignant brain tumor of the central nervous system in adults, accounting for nearly half of all malignant brain tumors [1]. Despite standard treatment consisting of maximal surgical resection followed by radiotherapy and concomitant temozolomide chemotherapy, the median overall survival remains only 14–16 months, with a five-year survival rate of less than 10% [2]. GBM’s invasive nature and therapeutic resistance stem from several factors: diffuse tumor infiltration, the restricted permeability of the blood–brain barrier (BBB), and the chemoresistance of GBM stem cells (GSCs). The latter is mediated by pathways such as Notch/Wnt signaling and O^6^-methylguanine-DNA methyltransferase (MGMT) overexpression [3,4]. Notably, recent single-cell sequencing studies have revealed that GBM harbors a highly immunosuppressive tumor microenvironment (TME), which constitutes an additional therapeutic barrier. The intricate immune escape mechanisms orchestrated by tumor-associated immune cells impair the efficacy of existing treatments [5,6,7].

Among the immunosuppressive components of the GBM microenvironment, tumor-associated macrophages (TAMs) are particularly prominent, comprising 30–50% of the tumor mass [8] (Figure 1). TAMs consist of both bone marrow-derived macrophages (BMDMs) and brain-resident microglia, which are the innate immune cells of the central nervous system and share many phenotypic and functional characteristics with macrophages. Their recruitment, differentiation, and function are tightly regulated by tumor-derived chemotactic factors such as CCL2 and CSF1 [9]. The immunosuppressive landscape of GBM exhibits a highly organized spatial structure: the tumor core is enriched with CD163^+^CD206^+^ M2-polarized BMDMs, which suppress T cell function through IL-10 and VEGF-A secretion while promoting angiogenesis [10]. In contrast, CX3CR1^+^ microglia at the tumor margin drive tumor invasion via matrix metalloproteinase (MMP)-mediated extracellular matrix remodeling [11]. This spatial and temporal heterogeneity highlights the important role of TAMs in linking tumor genetic diversity with immune evasion. Importantly, the functional plasticity of TAMs offers potential opportunities for therapeutic intervention. Targeting the CSF1R/STAT3 axis has been shown to reprogram TAMs toward a pro-inflammatory, tumoricidal phenotype, while blockade of the CD47-SIRPα axis restores macrophage-mediated phagocytosis, enhancing antitumor immunity [12]. In this review, we describe the role of TAMs in GBM immune evasion, focusing on key regulatory pathways such as CSF1R, STAT3, HIF-1α, and chemokine axes. Furthermore, we will summarize emerging TAM-targeted immunotherapeutic strategies, including TAM infiltration blockade, phenotypic reprogramming, selective depletion, and novel CAR-macrophage (CAR-M) therapies, with the aim of advancing precision immunotherapy for GBM.

## 2. Evolution of Macrophage Classification and Contemporary Understanding

### 2.1. The Classical M1/M2 Model and Its Limitations

The functional heterogeneity of macrophages was initially identified in studies on infection and tissue repair, leading to the establishment of the M1 (classically activated)/M2 (alternatively activated) polarization model [13]. M1 macrophages are activated by IFN-γ and LPS, which upregulate inducible nitric oxide synthase (iNOS) via STAT1 signaling, leading to the release of IL-6, TNF-α, and reactive oxygen species (ROS), thereby exerting potent pro-inflammatory and antitumor effects [14]. In contrast, M2 macrophages are induced by IL-4 and IL-13, which activate STAT6 signaling, promoting the expression of arginase 1 (Arg1) and facilitating tissue repair, angiogenesis, and immune suppression, as well as exhibiting tumor-promoting properties [15,16] (Figure 2). However, this binary polarization model fails to explain the dynamic plasticity of macrophages within solid tumors such as GBM [17]. Single-cell RNA sequencing studies have revealed that tumor-associated macrophages exist along a continuous phenotypic spectrum rather than distinct M1 or M2 subtypes. For instance, TAMs in the invasive margin of GBM co-express CD86 (an M1 marker) and CD206 (an M2 marker), indicating spatial and temporal plasticity shaped by distinct tumor microenvironments [18] (Figure 2).

### 2.2. Subclassification of TAMs in the Tumor Microenvironment

Using scRNA-seq and spatial omics, recent studies found unique TAM subpopulations and their relationships with GBM molecular subtypes, genetic origins, and therapeutic resistance. In the GL261 mouse model, an immunosuppressive TAM subgroup expressing CCL22, PD-L1, and CCL5 was found [19]. PD-L1 expression and traditional inflammatory markers distinguish immunosuppressive and proliferative macrophage populations in human GBM. Spatial transcriptomic analysis showed that some Mo-TAMs localize to hypoxic necrotic zones and produce adrenomedullin to disrupt endothelial junctions, causing leaky vasculature and hindering medication delivery [20]. 

Certain GBM backgrounds strongly influence the spatial and functional features of various TAM subgroups. Pro-inflammatory proliferative HGG-AM microglia enhance tumor growth via the ApoE–NLRP1 inflammasome axis in IDH1-wildtype/SETD2-mutant GBM [21]. In mesenchymal GBM, MARCO^+^ macrophages and CD163^+^HMOX1^+^ microglia are abundant, with the latter emitting IL-10 to deplete T cells [22]. Immune-resistant TAM populations, such as PD-L1^+^ and CD73 high macrophages, impact immune evasion indirectly by adjusting suppressive myeloid subgroup proportions [23]. 

These scRNA-seq and spatial omics investigations establish a new TAM classification system based on lineage origin, geographical localization, and functional status. The lineage axis includes embryonically derived microglia, monocyte-derived macrophages, and transdifferentiated subsets; the spatial axis shows TAMs’ heterogeneous localization in tumor compartments like perivascular regions, necrotic cores, and immune boundary zones; and the functional axis integrates immunosuppressive, pro-inflammatory, and metabolically reprogrammed states [24]. This three-dimensional model transcends the M1/M2 paradigm, identifies targetable TAM subsets for customized immunotherapeutic methods, and provides a conceptual basis for TAM-specific biomarker creation and validation.

### 2.3. Dual Origins of TAMs in GBM

In the central nervous system, tumor-associated macrophages primarily originate from two distinct cellular populations: microglia and BMDMs (Figure 1). Microglia originate from embryonic yolk sac progenitors, remain as resident CNS macrophages, and undergo self-renewal while exhibiting specific markers such as TMEM119 and P2RY12 [25,26]. During the early stages of tumorigenesis, microglia recognize damage-associated molecular patterns (DAMPs) via the TLR4/MyD88 signaling pathway, thereby initiating an antitumor immune response [27]. However, as GBM progresses, microglia undergo reprogramming, partially polarizing toward an M2-like phenotype, thereby promoting immune suppression and angiogenesis. In contrast, BMDMs primarily derive from peripheral monocytes that infiltrate the tumor core via CCR2-dependent pathways [28]. These cells are characterized by high CD45 and CD11b expression [29]. Under hypoxic conditions, BMDMs activate the HIF-1α/NF-κB pathway, leading to the secretion of IL-10 and Arg1, which further suppress antitumor immunity [30]. BMDM infiltration has been strongly associated with GBM therapy resistance, potentially by enhancing glioblastoma cell resistance to temozolomide (TMZ) and promoting vasculogenic mimicry (VM)—a process in which tumor cells form endothelium-independent, perfusable vascular networks [31]. Single-cell lineage tracing studies have demonstrated that as GBM progresses, the proportion of BMDMs gradually increases, eventually dominating the TAM population in late-stage GBM, which correlates with worse patient prognosis [32].

## 3. Roles of TAMs in GBM

### 3.1. TAMs and Immune Suppression

GBM is traditionally recognized as an “immune-cold” tumor, characterized by a profoundly immunosuppressive microenvironment. TAMs are key regulators in maintaining this immunosuppressive state [33]. First, TAMs exert direct immunosuppressive effects by expressing high levels of checkpoint molecules such as PD-L1 and B7-H4, which interact with PD-1 and CTLA-4 on T cells, thereby inhibiting effector T cell proliferation and cytotoxic function [9,34,35,36]. Furthermore, TAMs secrete multiple immunosuppressive factors, including TGF-β, IL-10, and CCL2, which attenuate antitumor immunity via distinct mechanisms. Specifically, TGF-β downregulates the expression of genes associated with T cells and natural killer (NK) cells, while IL-10 impairs dendritic cell antigen presentation, ultimately reducing CD8^+^ T cell activity [37]. Additionally, TAMs recruit regulatory T cells (Tregs) and myeloid-derived suppressor cells (MDSCs) through CCL2 secretion, establishing a widespread immunosuppressive network [38] (Figure 3). Beyond their immunosuppressive roles, TAMs regulate metabolic pathways to support GBM immune evasion. They deplete essential amino acids such as tryptophan and arginine, limiting T cell activation. AHR activation in TAMs, driven by GBM-secreted tryptophan metabolites like TDO2 and IDO1, upregulates CCR2 and Krüppel-like factor 4 (KLF4), which suppress NF-κB signaling and drive M2-like polarization, further enhancing immunosuppression [39]. The co-expression of CD73 and CD39 in TAMs facilitates the conversion of ATP into adenosine, which inhibits CD8^+^ T cell function and supports tumor immune escape [40,41]. These mechanisms collectively contribute to GBM’s resistance to immune checkpoint inhibitors, making it one of the most challenging solid tumors in clinical practice.

### 3.2. TAMs and Tumor Cell Interactions

Beyond TAM immunosuppressive functions, TAMs drive GBM progression through direct interactions with tumor cells. One critical mechanism involves TAM-derived extracellular vesicles (EVs), which transfer immunosuppressive proteins (CD73, CD39, PD-L1, and CTLA-4) to promote an immune-evading TME [42]. Additionally, TAMs support tumor growth through the secretion of pro-angiogenic factors such as vascular endothelial growth factor (VEGF), fibroblast growth factor (FGF), and CXCL2, contributing to neovascularization and enhancing nutrient supply to GBM cells. Additionally, TAMs upregulate extracellular matrix (ECM)-degrading enzymes, including MMP-9, MMP-2, and MMP-14, facilitating basement membrane degradation and neovascularization [23] (Figure 3). These newly formed blood vessels not only provide oxygen and nutrients to tumor cells but also exhibit structural abnormalities that limit chemotherapy drug delivery, thereby exacerbating therapeutic resistance [43]. TAMs form a paracrine signaling loop with glioma stem-like cells, where TAM-derived cytokines (TGF-β and IL-6) activate STAT3 and NF-κB signaling. This process enhances GSC self-renewal, stemness, and resistance to radio- and chemotherapy [44]. Conversely, GSCs secrete chemokines such as CXCL12 and CCL5, recruiting M2-polarized macrophages and establishing a positive feedback loop that accelerates GBM progression [45].

Another key function of TAMs in GBM progression is their regulation of phagocytosis. TAMs expressing the scavenger receptor MARCO have been identified as pro-tumorigenic in IDH1-wildtype GBM, where they suppress TNF-α signaling via NF-κB and impair antigen presentation, limiting antitumor immune responses [46]. TAMs also modulate phagocytosis via the CD47-SIRPα axis, wherein GBM cells upregulate CD47 to evade immune clearance [47]. In preclinical models, targeting CD47 with shRNA reprogrammed M2-like TAMs to an M1 phenotype and enhanced tumor cell phagocytosis [48]. Interestingly, a subset of IFN-γ-secreting CD169^+^ TAMs has been found to enhance phagocytosis and promote antigen-specific T cell responses by recognizing apoptotic tumor cell ligands [48,49]. These findings suggest that manipulating TAM-mediated phagocytosis could be a promising therapeutic strategy in GBM.

In addition to TAM–GSC interactions, a recent study uncovered a distinct immunosuppressive axis in IDH–wild-type glioblastoma, involving early-stage myeloid-derived suppressor cells (E-MDSCs) and stem-like glioblastoma cells. These E-MDSCs were found to colocalize with GSCs specifically within pseudopalisading necrotic regions, a histological hallmark of aggressive GBM. Mechanistically, GSCs secrete chemokines such as CXCL2 to recruit E-MDSCs, which in turn release fibroblast growth factor 11 (FGF11). FGF11 activates the FGFR1 signaling pathway in adjacent tumor cells, forming a pro-tumorigenic feedback loop that promotes tumor proliferation and stemness. Notably, this MDSC–GSC axis functions independently of classical TAM-driven pathways such as IL-6/STAT3, and its presence correlates with poor prognosis in IDH-WT GBM. These findings suggest that E-MDSCs and their molecular interactions with GSCs represent a novel therapeutic target, particularly in molecular subtypes of GBM characterized by high immune suppression and treatment resistance [50].

### 3.3. The Role of TAMs in Therapeutic Resistance

TAMs critically mediate resistance to chemotherapy, radiotherapy, and immunotherapy in GBM. In the context of chemotherapy resistance, the cytotoxicity of TMZ primarily relies on DNA damage. However, M2-like TAMs enhance tumor cell survival by secreting IL-6 and TGF-β, which activate DNA repair pathways such as MGMT and PI3K/Akt/STAT3, leading to increased TMZ resistance [51]. Additionally, TAMs secrete growth differentiation factor 15 (GDF15), which promotes tumor cell adaptation to oxidative stress and chemotherapeutic toxicity. With respect to radiotherapy resistance, radiation-induced hypoxia stimulates the CCL2/CCR2 axis, promoting monocyte infiltration and driving M2 polarization of TAMs. This shift leads to the secretion of VEGF and TGF-β, which facilitate vascular remodeling and enhance DNA repair mechanisms, ultimately conferring radioresistance [52]. CCR2 antagonists such as PF-04136309 have been shown to reduce tumor-associated macrophage infiltration and enhance antitumor immunity in pancreatic ductal adenocarcinoma, leading to improved tumor control in combination with chemotherapy [53]. Furthermore, TAMs profoundly impair the efficacy of immune checkpoint inhibitors. The high abundance of TAMs in GBM synchronously upregulates PD-L1 and Galectin-9 while collaborating with Tregs and MDSCs to suppress CD8^+^ T cell function, severely limiting the clinical efficacy of PD-1/PD-L1 blockade [18]. In clinical trials, monotherapy with PD-1 inhibitors such as nivolumab failed to yield significant survival benefits in patients with recurrent GBM. The CheckMate 143 trial reported a median overall survival (OS) of 9.8 months (95% CI, 8.2–11.8) in the nivolumab group, compared to 10.0 months (95% CI, 9.0–11.8) in the bevacizumab group, with no statistically significant difference (HR = 1.04; 95% CI, 0.83–1.30; *p* = 0.76). Additionally, the objective response rate (ORR) in the nivolumab group was only 7.8% (95% CI, 4.1–13.3%), markedly lower than the 23.1% (95% CI, 16.7–30.5%) observed in the bevacizumab group [54]. A complementary neoadjuvant cohort in surgically accessible recurrent GBM (NCT02852655) showed clear pharmacodynamic engagement with pembrolizumab—suppression of cell-cycle programs and induction of T cell/interferon signatures in resected tumors—yet did not confirm a survival benefit (pooled PFS-6, 19.5% [95% CI, 9.29–41.2%]) [55]. These findings indicate that PD-1 inhibitors alone fail to overcome the immunosuppressive GBM microenvironment, thereby limiting therapeutic efficacy. Beyond monotherapy, the recurrent-GBM cohort of the phase 2 LEAP-005 study reported that lenvatinib plus pembrolizumab achieved an ORR of 20% (95% CI, 13–29%), with median PFS of 3.0 months and median OS of 8.6 months, alongside grade 3–5 treatment-related adverse events in 41% (including two treatment-related deaths), indicating activity in a subset but modest survival gains and substantial toxicity [56]. In parallel, combinatorial strategies involving TAM-targeting therapies, such as CSF1R inhibitors (e.g., BLZ945) or CD47-SIRPα blockade, may enhance the antitumor effects of PD-1 inhibition, and such approaches are currently under clinical investigation (NCT02829723) [57]. These findings highlight the necessity of combinatorial approaches targeting both TAMs and immune checkpoints to enhance therapeutic efficacy in GBM.

### 3.4. TAMs and Autophagic Regulation

Autophagy in TAMs serves dual roles: it maintains macrophage viability and metabolic fitness in hypoxic/acidic niches and shapes their immunoregulatory output. By selective mitophagy and redox control, autophagy can limit inflammasome activity and favor M2-like polarization, increasing IL-10, TGF-β, and VEGF while restraining antigen presentation. These changes feed forward to tumor cells by (i) EV cargo remodeling (e.g., immunoregulatory proteins and microRNAs that blunt cytotoxic immunity and enhance DNA-repair/anti-apoptotic pathways), and (ii) paracrine cytokine signaling that induces autophagy in GBM cells, thereby buffering TMZ- and RT-induced stress [57]. Conversely, tumor-cell autophagy releases DAMPs (e.g., HMGB1, ATP/adenosine) that re-educate macrophages and stabilize M2 programs, creating a self-reinforcing autophagy–immunosuppression circuit [58]. Recent evidence further suggests that tumor-intrinsic regulators of autophagy can indirectly modulate TAM function: for example, inhibition of heparanase (HPSE) by the small molecule RDS 3337 in U87 glioblastoma cells was shown to block autophagic flux (LC3-II and p62/SQSTM1 accumulation) and trigger apoptosis (caspase-3 activation and PARP1 cleavage), highlighting how disrupting tumor-cell autophagy can rewire cell death pathways and potentially reshape the immune microenvironment [59]. Therapeutically, context-specific autophagy modulation is key: inhibiting tumor-cell autophagy can sensitize to TMZ/RT, whereas tuning (rather than globally suppressing) TAM autophagy may prevent M2 skewing without compromising macrophage survival. Conceptually, dual-targeting strategies—pairing autophagy inhibitors with CSF1R/STAT3 or CXCR4 blockade—could simultaneously destabilize TAM-supported GSC niches and restore treatment sensitivity, warranting biomarker-guided trials [60].

## 4. Core Molecular Drivers and Signaling Pathways Governing TAM Regulation

Within the GBM tumor microenvironment, TAMs are abundantly infiltrated and critically sustain immunosuppression, profoundly influencing tumor proliferation, angiogenesis, and therapeutic resistance. The recruitment, survival, and functional polarization of TAMs are tightly regulated by key signaling axes, including the CSF1/CSF1R/IL-34 axis, chemokine networks (CX3CL1/CX3CR1, CCL2/CCR2, and CXCL12/CXCR4), and core pathways such as STAT3, NF-κB, PI3K/Akt, and HIF-1α [9]. These pathways collectively drive the M2-like immunosuppressive phenotype of TAMs, fostering tumor survival and progression, thereby representing promising therapeutic targets.

Mechanistically, CSF1/IL-34–CSF1R activates PI3K/Akt–mTOR and ERK in macrophages, enforcing an FAO/OXPHOS-biased, IRF4/PPARγ program that raises IL-10, VEGF-A, and PD-L1 while suppressing antigen presentation—thereby promoting angiogenesis and limiting cytotoxic immunity [44]; glioblastoma stem cells supply CSF1/IL-34 and trophic cues that stabilize these CSF1R-dependent states in perivascular/hypoxic niches. Persistent IL-6/OSM/TGF-β-JAK/STAT3 signaling further imprints an M2-like identity and, via paracrine loops, activates STAT3 in GSCs to reinforce stemness/mesenchymal programs and TMZ/RT tolerance. PI3K/Akt in TAMs integrates CSF1R/IGF-1R/TLR inputs to drive mTORC1-linked biosynthesis and secretion of IL-6, TGF-β, and pro-angiogenic factors, which in turn activate survival/DNA-repair pathways in adjacent tumor cells and generate leaky neovasculature that impairs drug delivery. Under hypoxia, HIF-1α in TAMs induces ARG1, VEGF, and MMPs, and elevates CXCL12, establishing CXCL12–CXCR4 gradients that recruit CCR2+ monocytes and retain GSCs within immune-exclusive niches while fostering adenosine-rich, T cell-suppressive microenvironments. Collectively, these axes cooperate to polarize TAMs, stabilize GSC niches, and amplify therapy tolerance, providing a mechanistic rationale for co-targeting CSF1R/PI3K–Akt or HIF-1α/CXCR4 alongside radiochemotherapy and immune checkpoint blockade.

### 4.1. The CSF1R Axis: TAM Polarization and Immune Suppression

GBM cells drive the polarization of microglia and BMDMs toward an M2-like immunosuppressive phenotype by overexpressing CSF1 or IL-34, thereby sustaining their pro-angiogenic and immunosuppressive functions. Preclinical studies have demonstrated that CSF1R inhibitors, such as PLX3397 and BLZ945, significantly reduce TAM infiltration in murine GBM models [61]. However, Quail et al. found that while BLZ945 could transiently suppress tumor growth, resistance ultimately emerged due to compensatory activation of the insulin-like growth factor 1 receptor (IGF-1R)/PI3K signaling pathway. Further experiments confirmed that the combined inhibition of IGF-1R (e.g., with linsitinib) or PI3K (e.g., with buparlisib) significantly enhanced the antitumor effects of BLZ945 and prolonged survival in tumor-bearing mice [62].

In a phase I/II clinical trial (NCT02829723), BLZ945, either as monotherapy or in combination with the PD-1 inhibitor spartalizumab (400 mg IV every four weeks), demonstrated partial antitumor activity in patients with recurrent GBM. RNA sequencing revealed that the combination therapy upregulated interferon-γ pathway genes (e.g., CXCL9/10) and HLA molecule expression, suggesting a reprogramming of the immune microenvironment [57]. However, the clinical efficacy of CSF1R inhibitors as monotherapy remains limited. For instance, in a study involving recurrent GBM patients, PLX3397 achieved a six-month progression-free survival (PFS6) rate of only 8.6%, with no objective responses (ORR = 0%). Despite its ability to effectively penetrate the blood–brain barrier (with a tumor tissue concentration reaching 5500 ng/g), PLX3397 did not significantly prolong overall survival [63]. Currently, multiple clinical trials are investigating combination strategies involving CSF1R inhibitors and PD-1/PD-L1 inhibitors (e.g., pembrolizumab) [64]. These approaches aim to overcome the inherent resistance of GBM by simultaneously disrupting the immunosuppressive microenvironment and enhancing T cell infiltration.

### 4.2. Chemokine Axes: Recruitment and Functional Programming of TAMs

Within the GBM microenvironment, the chemokine axes play a pivotal role in the recruitment and functional reprogramming of TAMs. Studies have identified the CCL2/CCR2 axis as the primary pathway mediating the infiltration of bone marrow-derived monocytes into GBM [65]. GBM cells and tumor-associated endothelial cells secrete CCL2, which binds to CCR2 on circulating CX3CR1^+^CCR2^+^ monocytes, facilitating their transmigration across the blood–brain barrier into the tumor microenvironment [38]. Once within the tumor, these BMDMs differentiate into CX3CR1^+^CCR2^−^ TAMs and acquire an M2-like immunosuppressive phenotype, promoting immune evasion. Notably, BMDMs predominantly localize around tumor vasculature, whereas microglia are more concentrated in peripheral tumor regions [65,66]. Gene knockout experiments further demonstrated that CCL2 inhibition reduces BMDM recruitment, delays tumor progression, and prolongs survival in murine models, suggesting that the CCL2/CCR2 axis is a potential therapeutic target for GBM immunotherapy [67].

In addition to the CCL2/CCR2 axis, the CX3CL1/CX3CR1 axis also plays a role in GBM progression. Neurons and tumor cells express CX3CL1, which interacts with CX3CR1 on microglia, promoting tumor cell invasion and microenvironment modulation [67]. Another key pathway is the CXCL12/CXCR4 axis, which is predominantly upregulated in hypoxic tumor regions. CXCL12 binding to CXCR4 promotes aberrant angiogenesis and maintains the immunosuppressive phenotype of TAMs via the HIF-1α pathway [68]. Preclinical studies have shown that CXCR4 antagonists, such as AMD3100, in combination with PD-1 inhibitors, can effectively overcome the immunosuppressive tumor microenvironment and enhance antitumor immune responses. In the GL261 murine GBM model, combination immunotherapy with anti-CXCR4 and anti-PD-1 significantly reduced CD11b^+^ macrophage and monocytic MDSC infiltration while lowering the proportion of CD4^+^FoxP3^+^ Treg cells, thereby enhancing CD8^+^ tumor-infiltrating lymphocyte (TIL) cytotoxicity. Additionally, the combination therapy induced a robust IFN-γ and TNF-α-driven inflammatory response and promoted immune memory formation, leading to a significant survival benefit (*p* < 0.01). These findings support CXCR4 as a potential target for GBM immunotherapy and provide a rationale for future clinical trials [69].

Within the GBM microenvironment, CXCR4 facilitates the immunosuppressive and therapy-resistant phenotype of GBM by regulating SDF-1α (CXCL12)-mediated CD11b^+^ myeloid cell migration [70]. The CXCR4 antagonist Plerixafor effectively blocks this pathway, reducing the infiltration of TAMs and MDSCs, thereby enhancing antitumor immunity. Currently, a phase II clinical trial (NCT03746080) is evaluating the efficacy of Plerixafor in combination with whole-brain radiotherapy (WBRT) in newly diagnosed GBM patients. Although this regimen significantly improved six-month progression-free survival (PFS6 = 91.7%), it failed to prolong median overall survival (mOS = 15.11 months). Moreover, WBRT-induced cognitive decline was observed in 80% of patients at six months compared to baseline [71]. These findings suggest that the role of CXCR4 antagonists in GBM immunotherapy requires further optimization, potentially necessitating combination strategies with additional immunomodulatory approaches.

### 4.3. Immunoregulatory Signaling Pathways

Both STAT3 and HIF-1α play critical regulatory roles in maintaining the M2 immunosuppressive phenotype of TAMs and contribute to immune evasion in glioblastoma (Figure 3). Specifically, GBM cells secrete IL-6, which persistently activates the JAK2/STAT3 signaling cascade, while TGF-β may synergistically enhance this immunosuppressive effect, leading to elevated expression of IL-10, VEGF, and CD163 in TAMs. Consequently, TAMs exhibit enhanced immunosuppression and pro-angiogenic functions [72,73]. Notably, macrophage polarization and functional maintenance depend on a range of shared or specific signaling pathways that are crucial in inflammation, tumor immunity, and metabolic regulation.

During M1/M2 macrophage polarization, STAT3 and STAT1 display opposing roles: STAT3 is predominantly associated with the M2-like immunosuppressive state, whereas STAT1 is more inclined to drive the proinflammatory M1 phenotype. In GBM, the tumor cells and their secreted factors upregulate STAT3 activity in macrophages, thereby suppressing proinflammatory gene transcription and bolstering immune evasion. Therefore, inhibiting STAT3 can partially reverse the immunosuppressive properties of TAMs. For instance, in a phase I clinical trial (NCT01904123) [74], the STAT3 inhibitor WP1066 reduced p-STAT3 levels in peripheral blood but did not significantly affect FoxP3^+^ Tregs, and changes in CD8^+^IFN-γ^+^ T cells were inconsistent. Although WP1066 demonstrated biological activity, all patients experienced radiologically confirmed progressive disease (PD) with a six-month progression-free survival (PFS6) of 0%, indicating that WP1066 monotherapy failed to confer clinical benefit in GBM [74]. One possible explanation is that GBM-specific effector T cells may have trafficked to the tumor microenvironment or were sequestered in the bone marrow, thereby reducing their presence in peripheral circulation.

In addition to the JAK/STAT3 axis, the NF-κB signaling pathway is also a key regulator of macrophage function. NF-κB is activated by proinflammatory factors such as TNF-α and IL-1 and, in the tumor microenvironment, can drive both proinflammatory responses and, under certain conditions, be reprogrammed to support tumor cell survival and proliferation [75]. In GBM, NF-κB activation may further enhance the immunosuppressive capacity of M2-type TAMs and promote VEGF-mediated angiogenesis. Hence, NF-κB inhibitors may hold promise for TAM-targeted therapy [76].

Furthermore, the PI3K/Akt pathway governs cell proliferation, survival, and metabolic regulation. Studies have shown that the protumoral effects of M2 macrophages often coincide with PI3K/Akt activation, leading to increased IL-6 and TGF-β secretion, which in turn drives tumor cell resistance and enhances vascular permeability [77]. Consequently, blocking PI3K/Akt signaling may serve as a potential strategy to modulate macrophage polarization and limit tumor metastasis and therapeutic resistance.

In GBM tissue, a hypoxic niche is common due to the tumor’s high proliferation rate and aberrant vascular architecture. HIF-1α is stabilized under hypoxic conditions and induces the upregulation of VEGF, ARG1, and MMP9, thereby reinforcing the immunosuppressive functions of TAMs and facilitating tumor invasion [78]. In addition, lactate generated by tumor metabolism can further drive M2 polarization and enhance the secretion of immunosuppressive factors. Therefore, targeting HIF-1α or reducing lactate accumulation may mitigate TAM-mediated immunosuppression and promote the shift toward an M1 phenotype [79]. Notably, GBM patients with elevated HIF-1α levels exhibit poorer PFS and OS. The HIF-1α inhibitor PX-478 reduces VEGF and MMP9 expression in murine GBM models and impairs M2-type TAM-mediated angiogenesis (*p* < 0.001) [80].

## 5. Immunotherapeutic Strategies Targeting TAMs

Recent immunotherapeutic research has focused on blocking TAM recruitment, reprogramming their phenotype, selectively depleting them, and enhancing their function to combat GBM’s invasive and recurrent nature. Leveraging the insights gained from key molecular axes and signaling pathways, recent immunotherapeutic research focusing on “blocking TAM recruitment, reprogramming their phenotype, selective depletion, and functional enhancement” provides new avenues to combat the invasive and highly recurrent nature of GBM. Although these strategies have demonstrated certain synergistic and tumor-suppressive potentials in preclinical models, clinical applications remain challenging due to concerns regarding immune homeostasis disruption and drug toxicity.

### 5.1. Blocking TAM Infiltration

A fundamental approach involves curtailing the excessive infiltration of TAMs into tumor sites. Continuous recruitment of peripheral monocytes and certain microglia occurs under the influence of diverse chemokine pathways such as CCL2/CCR2, CX3CL1/CX3CR1, and CXCL12/CXCR4, driving their migration into GBM [81]. Accordingly, specific antagonists or molecular disruptors can reduce the abnormal infiltration of macrophages and diminish their immunosuppressive effects. However, these chemokine axes also play pivotal roles in normal immune homeostasis and inflammatory responses. Therefore, it is crucial to strike a balance between achieving antitumor efficacy and minimizing disruption of normal immune functions (Figure 3).

### 5.2. Reprogramming the TAM Immunophenotype

Reprogramming TAMs from an immunosuppressive M2 phenotype to an antitumor M1 phenotype is another promising approach. Compared with the outright depletion of TAMs, this strategy not only attenuates immunosuppression but also enhances the antitumor immune response within the tumor microenvironment, thereby improving therapeutic outcomes (Figure 3).

CSF1R inhibitors (e.g., GW2580 [82], BLZ945 [83], and PLX3397 [62]) interfere with the CSF1/CSF1R axis, suppressing the survival of M2-like TAMs and inducing their shift toward a proinflammatory M1 phenotype. In addition, immunosuppressive factors such as IL-10 and TGF-β are instrumental in maintaining M2 polarization. Blocking their signaling pathways can promote macrophage-driven antigen presentation and T cell activation, augmenting antitumor immune responses [84]. Recently, Toll-like receptor (TLR) activation has also emerged as an effective reprogramming strategy for TAMs. For example, TLR agonists (e.g., CpG ODNs) can stimulate macrophages to release proinflammatory cytokines (including TNF-α and IL-12), thereby enhancing the immunological attack within the tumor microenvironment [85]. Moreover, because the STAT3 signaling pathway is pivotal for M2 polarization, STAT3 inhibitors have been shown to effectively drive macrophages toward the M1 phenotype and significantly inhibit tumor growth in GBM animal models [86].

Importantly, a novel target for macrophage reprogramming is the scavenger receptor MARCO (macrophage receptor with collagenous structure), which is highly expressed on TAMs in various solid tumors. Blocking MARCO with a monoclonal antibody (e.g., ED31) has been shown to reprogram TAMs toward an M1 phenotype without depleting them [87]. This reprogramming enhances the secretion of M1-type chemokines (e.g., CCL2, CCL3, CCL4, CCL5, CXCL1, CXCL10, and CCL22), which in turn promotes the infiltration of various immune cells, including CD8^+^ T cells, NK cells, and mature dendritic cells (cDCs), into the tumor microenvironment. Notably, MARCO blockade synergizes with anti-CTLA-4 therapy, but not anti-PD-1, to significantly improve tumor control and survival in murine tumor models [87]. These findings support MARCO as an alternative immune checkpoint within the TME and suggest that targeting MARCO may transform “cold” tumors into immunologically active “hot” tumors.

TAM reprogramming strategies show strong synergy when combined with immune checkpoint inhibitors targeting PD-1/PD-L1 or CTLA-4. M1 macrophages enhance T cell activation, and reprogramming TAMs helps alleviate the immunosuppressive microenvironment, improving checkpoint inhibitor efficacy. Furthermore, combining CSF1R or STAT3 inhibitors with radiotherapy, chemotherapy, or oncolytic virotherapy has also been shown to bolster antitumor immune responses and presents new possibilities for overcoming GBM resistance [88].

### 5.3. Selective Depletion of TAMs

For TAM subpopulations that dominate the immunosuppressive milieu, selective elimination or depletion represents a feasible strategy (Figure 3). Methods such as FasL-induced apoptosis or clodronate-loaded liposomes can specifically target macrophages, thereby rapidly reducing TAM abundance and inhibiting tumor growth [89]. However, because macrophages are indispensable for maintaining normal tissue homeostasis and innate immunity, excessive depletion may compromise the host’s defensive capabilities. Consequently, determining the precise scope and dosing for macrophage depletion remains a central challenge in translational applications.

### 5.4. Enhancing the Phagocytic Capacity of TAMs

GBM cells frequently overexpress CD47, which interacts with SIRPα on macrophages to deliver a “don’t eat me” inhibitory signal [90]. Blocking the CD47-SIRPα axis (e.g., with the anti-CD47 antibody Hu5F9-G4) restores the phagocytic activity of macrophages against GBM cells and can further enhance therapeutic efficacy when combined with temozolomide or immune checkpoint inhibitors [91] (Figure 3). Moreover, emerging CAR-macrophage (CAR-M) technology uses genetic engineering to confer robust tumor phagocytic activity in immunosuppressive environments, suggesting a new avenue for personalized cell therapy in GBM [92].

### 5.5. The Potential of CAR-M Cell Therapy

As knowledge of the profoundly immunosuppressive microenvironment in glioblastoma continues to expand, engineering macrophages to express chimeric antigen receptors (CARs) has emerged as a novel approach to overcoming current therapeutic barriers [93]. Chimeric antigen receptors (CARs) are synthetic receptors that combine the specificity of an antibody with the signaling domains of T cells, enabling the cell to target specific tumor antigens. Macrophages possess intrinsic tumor-tropic properties, enabling them to cross the blood–brain barrier under the guidance of multiple chemotactic signals and to function within severely hypoxic tumor regions. By incorporating a CAR domain on the macrophage surface that specifically recognizes key GBM antigens (e.g., EGFRvIII), CAR-M cells can not only detect and engulf tumor cells with high precision but also amplify adaptive immune responses through the release of inflammatory cytokines and the activation of antigen presentation [94,95]. Preclinical studies have provided initial evidence demonstrating the safety and potent antitumor activity of this strategy in cell-based models [95]. However, major questions remain regarding CAR-M cell persistence and functional stability in humans, the mitigation of potential off-target effects on normal brain tissue, and the optimal combination with other immunotherapies or chemoradiation to prevent tumor recurrence [96].

### 5.6. Nanotechnology and Cell Carrier Strategies

In efforts to more effectively modulate TAMs within the GBM microenvironment and enhance their antitumor functionality, emerging nanotechnologies and engineered cell carriers have shown considerable promise [97]. On one hand, agents such as CSF1R or STAT3 inhibitors, as well as nucleic acid interference molecules (e.g., siRNA, miRNA), can be encapsulated in brain-targeted nanoparticles or liposomes with ligand modifications and stimulus-responsive designs. These formulations increase drug accumulation at the tumor site while minimizing damage to normal tissue [98]. On the other hand, researchers can leverage the natural infiltration and phagocytic capabilities of macrophages by transforming them into drug carriers or gene delivery vehicles, enabling localized high-concentration release within GBM lesions and promoting the shift from M2 to M1 phenotypes [99]. Additionally, exosomes—critical mediators of tumor-immune cell communication—may be employed to deliver specific non-coding RNAs or proteins for precise regulation of TAM function [100]. Although traversing the BBB remains a significant technical obstacle, ongoing advances in materials science (e.g., hypoxia- or acidity-responsive nanocarriers) and physical modalities (e.g., focused ultrasound) are helping to achieve an optimal balance between “efficient delivery” and “safety,” thereby offering more transformative pathways for comprehensive immunotherapy in GBM [101].

## 6. Conclusions

Emerging therapies targeting TAMs, including CSF1R inhibitors, CAR-M cell therapy, and nanotechnology-based delivery systems, exhibit promising preclinical efficacy in GBM. However, clinical translation remains challenging due to GBM’s molecular heterogeneity, adaptive immunosuppressive remodeling, and the difficulty of achieving statistically significant outcomes in limited patient cohorts. Future strategies should integrate multidisciplinary advances in neuro-oncology, immunology, and bioengineering to optimize precision targeting. Multi-omics approaches (e.g., single-cell/spatial transcriptomics, metabolomics) will be critical to dissect TAM heterogeneity, identify actionable immune evasion mechanisms, and enable adaptive therapeutic regimens.

Additionally, next-generation strategies should incorporate TAM reprogramming, immune checkpoint blockade, and smart delivery platforms (e.g., hypoxia-responsive nanoparticles), augmented by AI-driven drug discovery and organoid-based predictive modeling. By constructing complex networks from large-scale multi-omics data, AI can identify disease-relevant molecular patterns and causal relationships. Applied to TAM biology, such integrative frameworks can classify macrophage subtypes beyond conventional lineage markers, identify spatially confined immunosuppressive programs within hypoxic or perivascular niches, and link them with patient outcomes. AI-assisted clustering and predictive modeling may further facilitate patient stratification according to TAM-driven immune states, enabling the development of personalized treatment algorithms. Additionally, coupling AI-based drug discovery pipelines with omics-defined TAM vulnerabilities could accelerate the identification of small molecules or biologics capable of reprogramming immunosuppressive TAM subsets, thus accelerating the rational design of novel agents specifically aimed at reprogramming immunosuppressive TAMs [102,103]. Moreover, integrating high-throughput omics with longitudinal clinical monitoring and iterative feedback from AI-based analytics may help refine lead compounds, predict resistance pathways, and guide real-time adaptation of treatment regimens. Such synergy between multi-omics profiling and AI-enhanced therapeutic discovery holds promise for more precise and personalized strategies to improve survival and quality of life for GBM patients.

## Figures and Tables

**Figure 1 cells-14-01458-f001:**
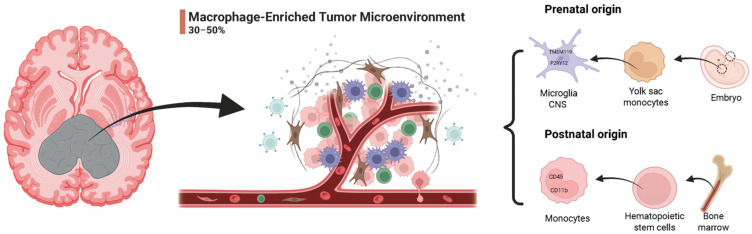
Macrophage-enriched tumor microenvironment and its origins in glioblastoma. The GBM microenvironment is heavily infiltrated by macrophages, which can constitute 30–50% of the total tumor mass. These TAMs accumulate around tumor vasculature and interact with tumor cells, playing critical roles in promoting tumor growth, immunosuppression, and resistance to therapy. TAMs originate from two distinct developmental pathways. Prenatally, they derive from yolk sac monocytes that migrate into the embryonic brain and differentiate into resident microglia, marked by TMEM119 and P2RY12 expression. Postnatally, circulating monocytes originating from hematopoietic stem cells in the bone marrow differentiate into macrophages upon entering the central nervous system, expressing markers such as CD45 and CD11b.

**Figure 2 cells-14-01458-f002:**
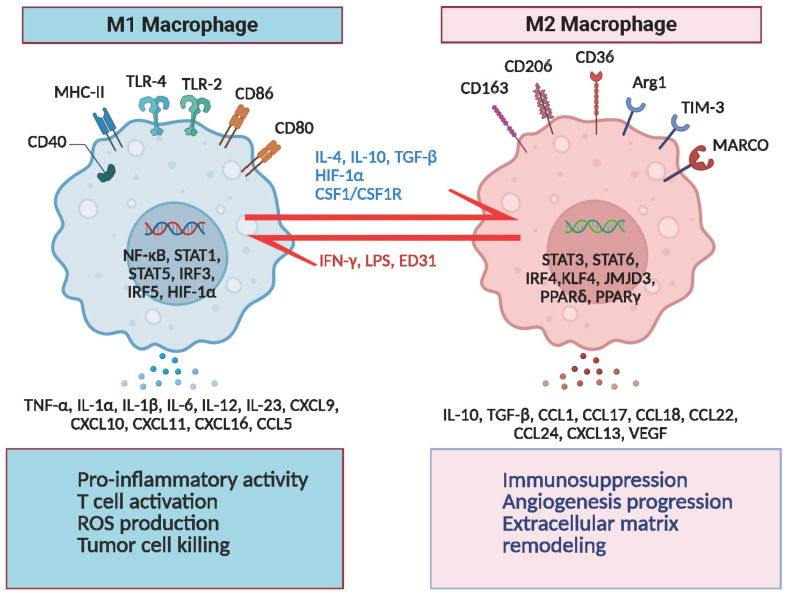
Polarization and functional characteristics of tumor-associated macrophages (TAMs). TAMs can be broadly polarized into two phenotypic states with distinct functions. M1 macrophages, activated by IFN-γ and LPS, express MHC-II, CD40, CD80, CD86, TLR4, and TLR2, and are regulated by NF-κB, STAT1, STAT5, IRF3, IRF5, and HIF-1α. They secrete pro-inflammatory cytokines (e.g., TNF-α, IL-1α, IL-1β, IL-6, IL-12, IL-23, CXCL9, CXCL10, CXCL11, CXCL16, and CCL5), promoting T cell activation, reactive oxygen species (ROS) production, and tumor cell killing. M2 macrophages, induced by IL-4, IL-10, TGF-β, CSF1, and HIF-1α, are regulated by STAT3, STAT6, IRF4, KLF4, JMJD3, PPARδ, and PPARγ, and express CD206, CD163, CD36, Arg1, TIM-3, and MARCO. They secrete IL-10, TGF-β, CCL1, CCL17, CCL18, CCL22, CCL24, CXCL13, and VEGF, contributing to immunosuppression, angiogenesis, and extracellular matrix remodeling. Targeting MARCO with monoclonal antibodies (e.g., ED31) can reprogram M2 macrophages toward a pro-inflammatory M1-like phenotype, thereby enhancing T cell infiltration, dendritic cell activation, and the efficacy of immune checkpoint blockade therapies.

**Figure 3 cells-14-01458-f003:**
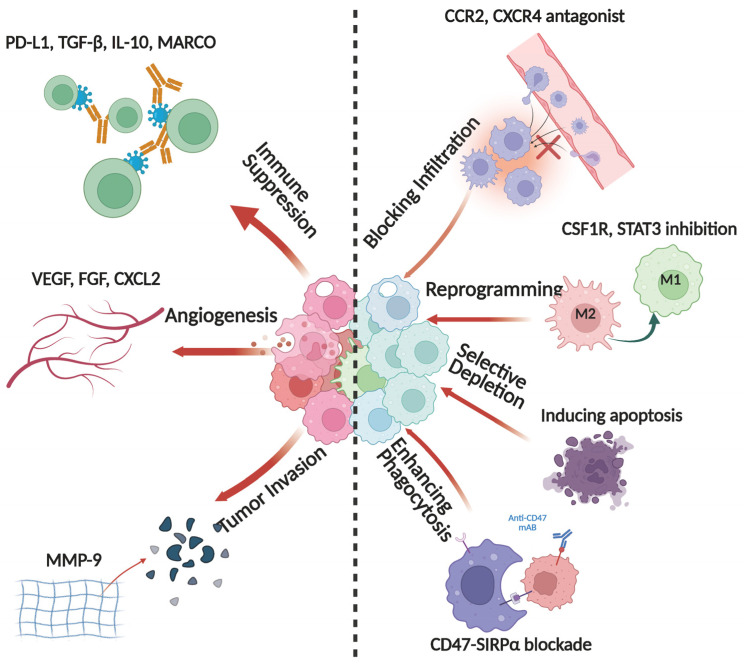
TAMs in the TME and therapeutic strategies. TAMs contribute to glioblastoma progression by creating an immunosuppressive tumor microenvironment. They inhibit T cell responses through PD-L1, B7-H4, TGF-β, IL-10, and CCL2, promote angiogenesis via VEGF and FGF, and enhance tumor invasion by secreting matrix metalloproteinases (e.g., MMP-2, MMP-9). MARCO suppresses the production of M1-type chemokines (e.g., CXCL9, CXCL10, CCL3, and CCL5), limiting immune cell infiltration into tumors. To counteract TAM-mediated immunosuppression, multiple therapeutic strategies are being explored. These include selective depletion of TAMs using CSF1R inhibitors, blocking monocyte recruitment via CCR2 or CXCR4 antagonists, and enhancing phagocytosis through CD47–SIRPα blockade or CAR-M therapy. Inhibiting MARCO (e.g., ED31) can reprogram TAMs toward an M1-like phenotype, restoring chemokine production and enhancing infiltration of cytotoxic T cells and mature dendritic cells, thereby synergizing with anti-CTLA-4 immunotherapy.

## Data Availability

Not applicable.

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
