# Peer review of "Tumor-Associated Macrophages in Glioblastoma: Mechanisms of Tumor Progression and Therapeutic Strategies"

_cells, 2025, doi:10.3390/cells14181458_

Round 1
Reviewer 1 Report
Comments and Suggestions for Authors
The manuscript reviews the literature on the interaction of macrophages with glioblastoma with the aim of outlining possible improvements in its therapy. It is relevant considering that glioblastoma (GBM) represents a formidable therapeutic challenge due to its profound immunosuppressive TME, which is predominantly shaped by innate immune mechanisms. The authors adequately describe the interaction of multiple mechanisms involved in the establishment of an immunosuppressive niche in this kind of tumor. The topics are well organized, following a logical progression. In fact, TAM repolarization, MDSC depletion, and NK cell activation, show promise in reshaping the GBM microenvironment and so contribute to new emerging therapies to glioblastoma, mainly if combined with other modalities.
I don'like very much the figures, mainly the figure 3 is quite complicate.
Author Response
Response to Reviewer 1
We sincerely appreciate the valuable feedback and insightful comments. Below, we address each comment in detail and describe the corresponding revisions made to the manuscript.
Comment 1:
The manuscript reviews the literature on the interaction of macrophages with glioblastoma with the aim of outlining possible improvements in its therapy. It is relevant considering that glioblastoma (GBM) represents a formidable therapeutic challenge due to its profound immunosuppressive TME, which is predominantly shaped by innate immune mechanisms. The authors adequately describe the interaction of multiple mechanisms involved in the establishment of an immunosuppressive niche in this kind of tumor. The topics are well organized, following a logical progression. In fact, TAM repolarization, MDSC depletion, and NK cell activation, show promise in reshaping the GBM microenvironment and so contribute to new emerging therapies to glioblastoma, mainly if combined with other modalities.
I don'like very much the figures, mainly the figure 3 is quite complicate.
Response:
We sincerely thank the reviewer for the constructive feedback and for the positive evaluation of our manuscript. We appreciate your comment regarding the complexity of Figure 3. In response, we have revised and simplified this figure to improve clarity and readability. We changed the labeling by using concise terms and reducing explanatory text, while maintaining the key mechanistic and therapeutic concepts. We hope that this modification addresses your concern and makes the figure more accessible to readers.

Reviewer 2 Report
Comments and Suggestions for Authors
The manuscript by Jianan Chen et al. provides a comprehensive and timely review of the role of tumor-associated macrophages (TAMs) in glioblastoma (GBM). It highlights their contribution to the immunosuppressive microenvironment, the mechanisms underlying their recruitment and polarization, and their potential as therapeutic targets. In addition, the discussion of emerging therapeutic strategies—such as CAR-macrophages, nanotechnologies, and exosome-based approaches—offers a forward-looking perspective that enhances the translational relevance of the review.
I have some suggestions :
The discussion of GBM heterogeneity and TAM plasticity as barriers to clinical success is highly relevant but could be further developed. Including more recent examples—such as trial failures with STAT3 inhibition or the limited survival benefit from PD-1 blockade—would concretely illustrate how adaptive TAM responses undermine immunotherapy efficacy.
The mention of AI-driven drug discovery and multi-omics integration is forward-looking but underdeveloped. The authors could elaborate on how AI-based approaches and spatial/single-cell multi-omics could identify TAM subtypes, stratify patients, or uncover novel TAM-targetable biomarkers.
Expanding on how pathways such as CSF1R/STAT3, PI3K/Akt, or HIF-1α specifically regulate macrophage phenotype, therapy resistance, and crosstalk with glioblastoma stem cells would strengthen the mechanistic depth. The section on TAM-mediated autophagy (Section 3.4) is particularly promising and could be expanded by integrating how autophagic regulation in TAMs intersects with tumor cell survival mechanisms.
A graphical abstract or schematic summarizing TAM roles and therapeutic strategies would improve clarity and accessibility.
For additional molecular insight: Recent studies have highlighted that targeting intracellular regulators can modulate the balance between apoptosis and autophagy in GBM cells, ultimately impacting immune responsiveness. For instance, an article from Cells of 2023 demonstrated that a novel heparanase inhibitor influences cell death pathways in U87 glioblastoma cells, suggesting that tumor-intrinsic processes may indirectly regulate TAM function and the surrounding microenvironment. Integrating this type of evidence would broaden the discussion by connecting TAM biology to tumor-intrinsic vulnerabilities.
Author Response
Response to Reviewer 2
We sincerely appreciate the valuable feedback and insightful comments. Below, we address each comment in detail and describe the corresponding revisions made to the manuscript.
Comment 1:
The discussion of GBM heterogeneity and TAM plasticity as barriers to clinical success is highly relevant but could be further developed. Including more recent examples—such as trial failures with STAT3 inhibition or the limited survival benefit from PD-1 blockade—would concretely illustrate how adaptive TAM responses undermine immunotherapy efficacy.
Response:
We thank the reviewer for this helpful suggestion. In the revised manuscript, we incorporated two recent clinical studies that exemplify the limited clinical benefit of PD-1–based approaches in GBM despite biological engagement of the immune axis (Line 219-235): (i) the stage-2 neoadjuvant pembrolizumab cohort in surgically accessible recurrent GBM (Nat Commun 2024; NCT02852655), which demonstrated clear pharmacodynamic effects (suppressed cell-cycle programs and increased T-cell/interferon signatures) but did not confirm a survival benefit (pooled PFS-6 = 19.5%); and (ii) the recurrent-GBM cohort of LEAP-005 (Cancer 2025), where lenvatinib plus pembrolizumab achieved an ORR of 20% yet yielded a median PFS of 3.0 months and OS of 8.6 months with notable grade 3–5 toxicities. These additions strengthen our discussion of how GBM heterogeneity and TAM plasticity undermine immune checkpoint inhibitors efficacy and support the need for TAM-informed combinations. Regarding STAT3 inhibition, an updated literature search did not identify additional peer-reviewed GBM trials beyond the well-known phase I WP1066 study, which showed pharmacodynamic p-STAT3 suppression without objective responses; this trial is already cited and discussed in our manuscript.
Comment 2:
The mention of AI-driven drug discovery and multi-omics integration is forward-looking but underdeveloped. The authors could elaborate on how AI-based approaches and spatial/single-cell multi-omics could identify TAM subtypes, stratify patients, or uncover novel TAM-targetable biomarkers.
Response:
We thank the reviewer for this insightful suggestion. We have expanded the discussion to elaborate on the synergistic role of AI-based approaches and multi-omics integration in TAM research. We now highlight how single-cell and spatial transcriptomics, when combined with AI-driven analytics, can identify novel TAM subtypes, reveal spatially organized immunosuppressive niches, and stratify patients according to TAM-related signatures. We have added these points in the Future Perspectives section (Line 531-539) to outline their translational potential.
Comment 3:
Expanding on how pathways such as CSF1R/STAT3, PI3K/Akt, or HIF-1α specifically regulate macrophage phenotype, therapy resistance, and crosstalk with glioblastoma stem cells would strengthen the mechanistic depth. The section on TAM-mediated autophagy (Section 3.4) is particularly promising and could be expanded by integrating how autophagic regulation in TAMs intersects with tumor cell survival mechanisms.
Response:
We appreciate this helpful suggestion. In the revised manuscript, we have rewritten Section 3.4 to integrate how TAM autophagy maintains macrophage viability and M2-like programming under hypoxia/acidic stress, reshapes EV/cytokine outputs, and intersects with tumor-cell survival pathways to buffer TMZ/RT-induced stress, thereby contributing to therapeutic resistance (Line 242-263). In addition, we added a new paragraph in Section 4 (Line 274-290) that details how CSF1R/STAT3, PI3K/Akt, and HIF-1α regulate macrophage phenotype, including induction of IL-10/VEGF/PD-L1, suppression of antigen presentation, angiogenic and chemokine remodeling, and bi-directional crosstalk with GSCs that stabilizes stemness and mesenchymal programs.
Comment 4:
A graphical abstract or schematic summarizing TAM roles and therapeutic strategies would improve clarity and accessibility.
Response:
Thank you for this helpful suggestion. We have carefully revised Figure 3 into a schematic summary that addresses this point. The left panel illustrates the principal roles of TAMs in GBM, and the right panel outlines the corresponding therapeutic strategies.
Comment 5:
For additional molecular insight: Recent studies have highlighted that targeting intracellular regulators can modulate the balance between apoptosis and autophagy in GBM cells, ultimately impacting immune responsiveness. For instance, an article from Cells of 2023 demonstrated that a novel heparanase inhibitor influences cell death pathways in U87 glioblastoma cells, suggesting that tumor-intrinsic processes may indirectly regulate TAM function and the surrounding microenvironment. Integrating this type of evidence would broaden the discussion by connecting TAM biology to tumor-intrinsic vulnerabilities.
Response:
We appreciate the suggestion and have integrated Cells evidence on HPSE inhibition in U87 cells, which blocks autophagic flux and triggers apoptosis. We added a concise paragraph to Section 3.4 (Line 252-257) linking tumor-intrinsic autophagy control to TAM conditioning and cited the article.
We also updated the reference numbering to reflect the newly added citations, and all modifications in the revised manuscript have been highlighted for your review.

Round 2
Reviewer 2 Report
Comments and Suggestions for Authors
The authors have completed my requests